# ZERO-SHOT CLASSIFICATION REVEALS POTENTIAL POSITIVE SENTIMENT BIAS IN AFRICAN LANGUAGES TRANSLATIONS

**Saurav K. Aryal, Hrishav Sapkota & Howard Piroleau**
Department of Computer Science
Howard University
Washington, DC 20059, USA
{hrishav.sapkota,saurav.aryal,howard.piroleau}@bison.howard.edu

## ABSTRACT

Natural Language Processing research into African languages has been limited, with over 2000 languages still needing to be studied. We employ the AfriSenti-SemEval dataset, a recently released resource that provides annotated tweets across 13 African languages, for sentiment analysis to address this. However, given the persistent data limitations for specific languages, we translate each language to English and conduct zero-shot classification using a large BART model trained with three candidate labels: positive, neutral, and negative. Intriguingly, our findings indicate that all tweets are classified as positive. Further investigation into prediction probabilities reveals that translation technologies may exhibit a bias in translating African languages toward positive sentiments. This observation highlights the potential impact of translation tools on sentiment analysis and warrants further examination.

## 1 INTRODUCTION

Despite being the most resource-abundant continent, Africa still struggles (Abu-Zaid & Mahfouz-Agouza, 2021). Africa's development has been widely hindered due to climate and geopolitical issues that have prolonged for centuries (Collier & Gunning, 1999). Out of several sectors impacted by these issues, the lack of preservation and integration of native languages spoken across the continent goes largely unaddressed and unnoticed (Alexander, 2009). The lack of adequate research output and educational infrastructure further hinders Natural Language Processing (NLP) research for African languages. While supervised modeling for sentiment analysis across multiple African languages is ongoing (Aryal et al., 2023), these approaches are not viable for languages that lack annotated datasets. As such, this paper seeks to experiment with the potential of translating and using a large language model pre-trained in English to perform language-agnostic sentiment analysis.

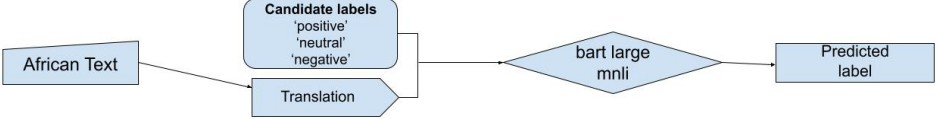

Figure 1: Proposed Modeling Approach

## 2 METHODOLOGY

A high-level diagram of our proposed approach can be seen in Figure 1. The approach was evaluated using datasets obtained from AfriSenti-SemEval, a corpus consisting of annotated tweets in various African languages (Yimam et al., 2020; Muhammad et al., 2022). Our analysis was performed on ten different languages, including Hausa (HA), Yoruba (YO), Igbo (IG), Amharic (AM), Algerian

Arabic (DZ), Swahili (SW), Kinyarwanda (KR), Twi (TWI), Mozambican Portuguese (PT), Xitsonga (Mozambique Dialect) (TS), and a combined dataset comprising all ten languages (ALL). Since all the datasets were collected from Twitter, we contend that the results of our analysis reflect real-world conditions. The translation process was conducted using the Google Translate API, considering potential transliteration and code-switching challenges. To ensure robustness, we employed two distinct methods of translation: a single-tweet approach and a word-by-word approach. Evaluations were carried out for both sentence-level and word-level translations, by language and across the entire set of languages. We opted not to conduct additional pre-processing besides removing URLs and numeric text.

Various models have been proposed for zero-shot classification. Among them, the BART model (Lewis et al., 2019) has shown promising results as a state-of-the-art benchmark for several zero-shot classification tasks (Chen et al., 2021; Tesfagergish et al., 2022; Gera et al., 2022). However, given the resource-intensive nature of training a BART model from scratch, we opted to utilize a pre-trained large BART model on Multi-Genre Natural Language Inference (MultiNLI) corpus, publicly available via Hugging Face (Wolf et al., 2020).

We obtained class probabilities by feeding the model with candidate labels, namely positive, neutral, and negative, and subsequently querying each translated tweet. Notably, our approach did not involve any training; thus we conducted our evaluation over the entire corpus. The relevant details regarding our evaluation corpus's sample sizes and class distribution are tabulated in Table 1. Furthermore, our experimental results led us to examine the probability distribution of each candidate label, which is presented in the same table.

Table 1: Sample size, Class Distribution, and Predicted Probabilities

| Lang | n | pos | neg | neu | Sentence | | | Word | | |
|------|---|-----|-----|-----|----------|----------|----------|----------|----------|----------|
| | | | | | P(pos) | P(neu) | P(neg) | P(pos) | P(neu) | P(neg) |
| HA | 16849 | 5574 | 5467 | 5808 | 0.82 ± 0.18 | 0.12 ± 0.12 | 0.06 ± 0.07 | 0.82 ± 0.18 | 0.12 ± 0.12 | 0.06 ± 0.07 |
| YO | 10612 | 4426 | 2315 | 3871 | 0.72 ± 0.21 | 0.18 ± 0.13 | 0.11 ± 0.09 | 0.73 ± 0.20 | 0.17 ± 0.12 | 0.10 ± 0.08 |
| IG | 12033 | 3644 | 3070 | 5319 | 0.76 ± 0.21 | 0.15 ± 0.13 | 0.09 ± 0.09 | 0.78 ± 0.20 | 0.14 ± 0.12 | 0.08 ± 0.08 |
| AM | 7481 | 1665 | 1936 | 3880 | 0.77 ± 0.19 | 0.15 ± 0.13 | 0.08 ± 0.08 | 0.75 ± 0.19 | 0.16 ± 0.12 | 0.09 ± 0.08 |
| DZ | 2065 | 522 | 1115 | 428 | 0.75 ± 0.21 | 0.16 ± 0.13 | 0.09 ± 0.08 | 0.76 ± 0.20 | 0.15 ± 0.12 | 0.09 ± 0.08 |
| SW | 2263 | 684 | 239 | 1340 | 0.71 ± 0. 19 | 0.18 ± 0.12 | 0.11 ± 0.08 | 0.71 ± 0.19 | 0.18 ± 0.12 | 0.11 ± 0.08 |
| KR | 4129 | 1124 | 1433 | 1572 | 0.74 ± 0.20 | 0.17 ± 0.13 | 0.09 ± 0.08 | 0.76 ± 0.19 | 0.16 ± 0.12 | 0.09 ± 0.08 |
| TWI | 3869 | 1827 | 1462 | 580 | 0.78 ± 0.19 | 0.14 ± 0.13 | 0.07 ± 0.07 | 0.79 ± 0.18 | 0.14 ± 0.13 | 0.07 ± 0.07 |
| PT | 3830 | 852 | 978 | 2000 | 0.81 ± 0.18 | 0.12 ± 0.12 | 0.07 ± 0.07 | 0.79 ± 0.19 | 0.14 ± 0.12 | 0.07 ± 0.08 |
| TS | 1007 | 480 | 356 | 171 | 0.74 ± 0.20 | 0.17 ± 0.13 | 0.09 ± 0.08 | 0.74 ± 0.20 | 0.17 ± 0.13 | 0.09 ± 0.08 |
| ALL | 64138 | 20798 | 18371 | 24969 | 0.77 ± 0.20 | 0.15 ± 0.13 | 0.08 ± 0.08 | 0.77 ± 0.19 | 0.14 ± 0.12 | 0.08 ± 0.08 |

## 3 RESULTS

Notably, the proposed approach yielded exclusively positive sentiment predictions for both translations and all languages. In light of the unsuccessful outcome of our attempt at zero-shot classification, we scrutinized the class probabilities generated by our approach. Upon analyzing the predicted probabilities within 1 standard deviation from the mean, as depicted in Table 1, we observed that the predicted probabilities were overwhelmingly positive across all languages. We surmise that this outcome may stem from two potential factors warranting further investigation. Firstly, translations may not convey sentiment consistently across languages, thereby leading to a loss of sentimental information. Additionally, given the unanimity of our findings, it is plausible that the translation models, the data utilized to train them, and the algorithms employed may harbor a bias toward data samples exhibiting positive sentiment.

## 4 CONCLUSION

Although our suggested method of utilizing English translations of African languages for zero-shot classification did not produce desired outcomes, our results indicate the necessity for further development of language translations capable of conveying sentiments between different languages. As the Afro-futurism movement continues to grow (Kim, 2017) and there is a greater demand for language research tailored to and conducted by Africans (Martinus & Abbott, 2019; Orife et al., 2020), current NLP researchers should acknowledge Africa's abundance of information and collaborate to enhance the progression of NLP for African and low-resourced languages.

URM STATEMENT

The authors acknowledge that all key authors of this work meet the URM criteria of ICLR 2023 Tiny Papers Track.

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
