# OpenReview forum: "Zero-Shot Classification Reveals Potential Positive Sentiment Bias in African Languages Translations"
_ICLR.cc/2023/TinyPapers — Submitted to Tiny Papers @ ICLR 2023_

### Official Review · Reviewer_BfnW · 2023-03-24

**Confidence:** 5

**Summary Of Contributions:**

The study employed zero-shot classification to analyze the sentiment of tweets in 13 African languages using the AfriSentiSemEval dataset. However, due to data limitations, the tweets were translated to English and analyzed using a BART model. The findings indicated that translation technologies may have a bias toward positive sentiments in African languages, highlighting the need for further investigation into the impact of translation tools on sentiment analysis

**Rating:**

High Potential (HP): a submission which meets the reviewing criteria and has potential to make an impact on the field

**Strengths And Weaknesses:**

- **Clarity:** Strength
- **Correctness:** Strength
- **Reproducibility:** Strength
- **Follows basic requirements:** Strength

**Suggested Changes:**

The research paper is commendable for its clarity and high quality. However, I think it would have been great if the AfriSentiSemEval dataset used in the study was discussed in more detail, including a link to where the dataset could be accessed. This would have allowed for further exploration of the dataset and added to the paper's overall value.

---

### Official Review · Reviewer_PTPk · 2023-04-02

**Confidence:** 5

**Summary Of Contributions:**

The authors performed zero-shot classification on Afrisenti 13 dataset using the BART model and this dataset was translated to english via google translate. The Afrisenti dataset contains all 3 sentiment labels but the zero-shot classification done by the authors resulted in only positive class.

**Rating:**

Great Start (GS): a submission which meets some of the reviewing criteria but has room for improvement

**Strengths And Weaknesses:**

Strengths

- The paper aims at exploring sentiment analysis in the African domain as this field is an under-explored one.
- The authors made an attempts to fill the gap of "data limitations" for this project.

Weaknesses
- The dataset was translated to english before the zero-shot classification was done, there is lack of clarity on the goal of translating the dataset to english was as zero shot classification was the Task C of Afrisenti Semeval shared task 12. The authors said to have translated to english because of dataset limitations but I believe data limitations is one of the reasons zero-shot learning was used.
- Translating African languages with google translate API is not the best option as the translation google translate gives for these languages are not always accurate.


**Suggested Changes:**

- Zero-shot classification could be done using an Afrocentric model(i.e models trained on African languages). This annuls the need to translate the dataset to english.
- If the reason for translating to english asides data limitation can be established, the authors can try human annotators or seq2seq Afrocentric language models for their translation.

---

### Meta-Review · Area_Chair_kFsp · 2023-04-07

**Recommendation:** Invite to present
**Confidence:** 3

**Metareview:**

This paper looks into zero-shot classification on Afrisenti 13 dataset using the BART model. In the analysis they observe that the Afrisenti dataset (which was translated by google api) contains all 3 sentiment labels but the zero-shot classification done by the authors resulted in only positive class.
The paper is well-written and well-presented. The problem the authors investigate is an interesting one and the reviewers agree it's valuable. One recommendation for the authors is to explore other translation options such as human annotators or Afrocentric seq2seq language models to shed light on this bias more in-depth.






**Summary:**

Analysis into bias in zero shot classification of African Languages Translations

**Comments And Feedback To The Authors:**

Overall, the paper's contribution to exploring sentiment analysis in the African domain is worthy but we suggest some improvements to enhance the research's impact.

**Reason For Not Giving A Higher Recommendation:**

The authors should provide more details about the evaluation to clarify questions of the reviewers.

**Reason For Not Giving A Lower Recommendation:**

The paper is well-presented, the hypothesis is well-supported, and the question is an interesting one for the field.

---

### Decision · Program_Chairs · 2023-04-09

Invite to present